# Streptococcal Toxic Shock Syndrome: Diagnostic and Therapeutic Approaches Incorporating Point-of-Care Antigen Testing—Case Series

**DOI:** 10.3390/clinpract15040070

**Published:** 2025-03-28

**Authors:** Peter Mihalov, Simona Kovalíková, Karol Laktiš, Matej Bendžala, Peter Sabaka

**Affiliations:** Department of Infectology and Geographical Medicine, Faculty of Medicine, Comenius University in Bratislava, 83101 Bratislava, Slovakia; mihalovpeter@gmail.com (P.M.); simkakovalikova@gmail.com (S.K.); laktis1@uniba.sk (K.L.); bendzala1@uniba.sk (M.B.)

**Keywords:** *Streptococcus pyogenes*, streptococcal toxic shock syndrome, tonsillitis, point-of-care test

## Abstract

Background: Streptococcal toxic shock syndrome (STSS) is a serious and potentially fatal complication of invasive Streptococcus pyogenes (Group A Streptococcus, GAS) infection, often stemming from severe soft tissue infections. While uncommon, tonsillitis can also lead to STSS, with lethality rates reported as high as 70%. Methods: We present three cases of patients diagnosed with tonsillitis who developed STSS. Point-of-care testing for GAS antigens was utilized to facilitate rapid diagnosis. Results: In all three cases, the characteristic clinical presentations, including scarlatiniform rash, strawberry tongue, and signs of shock with multi-organ dysfunction, were observed. Microbiological evidence confirmed ongoing GAS infections, and point-of-care testing for GAS antigens assisted in the diagnosis of tonsillitis in conjunction with STSS, enabling timely treatment interventions. Conclusions: Rapid diagnosis is crucial for the optimal management of STSS. The point-of-care testing for GAS may be useful for diagnosing STSS associated with tonsillitis.

## 1. Introduction

Streptococcal toxic shock syndrome (STSS) is a severe, rapidly progressive, and almost universally fatal condition if not treated promptly and correctly. The causative agent is *Streptococcus pyogenes* or Group A *Streptococcus* (GAS), according to the Lansfled classification, which produces streptococcal pyrogenic exotoxin with superantigen activity [1,2,3]. The development of clinical manifestations and complications of STSS is mediated by the interaction of streptococcal superantigens with the host immune system [4,5,6,7]. The disease is relatively rare, with a peak incidence in the elderly population. Other risk factors for the development of STSS include the presence of severe comorbidities (diabetes mellitus, immunodeficiency, and oncologic diseases), alcoholism, and impaired skin integrity [8]. The initial presentation is non-specific and often flu-like, with later development of maculopapular rash and shock with multi-organ dysfunction [9,10,11]. According to epidemiological studies, the lethality of STSS ranges from 30 to 70% [8]. It usually complicates soft tissue GAS infections. In rare cases, STSS has developed as a complication of tonsillitis or tonsillopharyngitis [12,13]. The cornerstone of appropriate management is the early administration of antibiotics, accompanied by supportive treatment of shock and surgical debridement if necessary. Early diagnosis is essential for appropriate management. This emphasises the importance of awareness of STSS, its signs and symptoms, and possible means of microbiological diagnosis. We present three cases of young patients with STSS, which developed as a complication of *Streptococcus pyogenes* tonsillitis, and demonstrate the usefulness of a point-of-care antigen test for *Streptococcus pyogenes* in the rapid diagnosis of STSS.

## 2. Results

### 2.1. Case 1

A 33-year-old obese (body mass index: 32.4 kg/m^2^) Caucasian female (ethnicity—Slovak, West Slavic) with a history of heroin abuse (buprenorphine/naloxone substitution therapy) presented to the Emergency Department of the Clinic of Infectious Diseases and Geographical Medicine, University Hospital in Bratislava, with a 3-day history of fever up to 41 °C, progressive weakness, malaise, pruritic rash and severe sore throat, and an approximately 6 h history of nausea, vomiting, and dyspnoea. The patient took azithromycin 500 mg orally 16 h before presentation, which was prescribed by a general practitioner. On presentation, the patient was agitated, hypotensive (60/30 mm of mercury), tachycardic (160 bpm), and tachypnoeic (36 breaths per minute). The patient tolerated only horizontal positions and any attempt to sit resulted in collapse. Physical examination revealed a generalised pink to light red confluent maculopapular rash with a sandpaper texture (Figure 1 and Figure 2), red swollen tonsils, dark red strawberry tongue, and crackles throughout the lungs. Laboratory investigations revealed elevated C-reactive protein (321 mg/L), interleukin-6 (996 pg/mL), procalcitonin (29.4 ng/mL), lactate (3.9 mmol/L), creatinine (280 umol/L), neutrophilia (24,700 cells/mL), plasma glucose (18.2 mmol/L), hyponatremia (123 mmol/L), and hypoxaemia (59 mm of mercury O_2_ in arterial blood). The results of laboratory tests, vital signs, and signs and symptoms at the time of presentation are shown in Table 1. Chest X-ray showed interstitial pulmonary oedema, and abdominal ultrasound showed hepatosplenomegaly. The oropharyngeal swab was antigen-positive for GAS using a rapid point-of-care test. Based on the clinical presentation and the positive GAS antigen, STSS was suspected. The patient was admitted to the intensive care unit. Empiric antimicrobial therapy (ceftriaxone 2 g every 12 h plus clindamycin 600 mg every 8 h), fluid resuscitation (30 mL of a balanced crystalloid solution per kilogram of body weight within the first 3 h), and vasopressor treatment with norepinephrine were started. Ceftriaxone was used instead of penicillin because of a history of penicillin allergy (rash). Due to hypoxaemia, the patient received oxygen via a face mask (FiO_2_ 60%). Hyperglycaemia was treated with continuous intravenous insulin. After 12 h of treatment, vital functions stabilised, and hypotension and tachycardia resolved. The patient was oliguric during the first 12 h (urine output 175 mL/12 h); however, diuresis was restored, and renal replacement therapy was not utilised. Oropharyngeal swabs were later cultured for *Streptococcus pyogenes*. Blood cultures taken at the time of presentation to the emergency department were negative. This may have been due to previous antibiotic therapy. Since *Streptococcus pyogenes* was isolated from a non-sterile site (the tonsils) rather than from a sterile site (with blood cultures returning negative), this case can only be classified as probable STSS. The cultivation of GAS from a sterile site is required to conclude a definitive diagnosis of STSS. On the second day of treatment, the patient’s vital signs were stable, procalcitonin, creatinine, neutrophilia, and lactate decreased, and urine output was restored. However, their sore throat became more prominent and their cervical lymph nodes were swollen and tender. Ultrasound of the neck showed enlarged vascularised lymph nodes along the mm. sternocleiodmastoidei. On the third day of treatment, norepinephrine therapy was stopped. The exanthema became paler and less visible. On day 8, respiratory insufficiency resolved, supplemental oxygen therapy was discontinued, and the patient was transferred from the ICU to the general ward. The patient however evolved palmar desquamation. Parenteral antibiotics were changed to oral antibiotics. Despite clinical stabilisation, hyperglycaemia did not resolve. Glycated haemoglobin was significantly elevated (11.2%), C-peptide was normal, and antibodies to glutamic acid decarboxylase were negative. Based on these findings, the patient was diagnosed with type 2 diabetes and treated with intensive insulin therapy. The patient was discharged on day 17. 

### 2.2. Case 2

A 20-year-old healthy Caucasian female (ethnicity—Slovak, West Slavic) presented to the Emergency Department of the Clinic of Infectious Diseases and Geographical Medicine, University Hospital in Bratislava, with a four-day history of fever up to 39 °C, severe sore throat, weakness, malaise, pruritic rash, nausea, vomiting, and diarrhoea. The patient was hypotensive (88/50 mm of mercury), tachycardic (137 beats per minute), and tachypnoeic (24 breaths per minute). Oxygen saturation measured by pulse oximetry was 92%. Physical examination revealed a generalised pink to light red maculopapular rash with a sandpaper texture. It was most prominent over the cubital region (Figure 3 and Figure 4). In the abdominal region, the exanthema was papular and pale (Figure 5). The tonsils were swollen and erythematous with purulent exudate. Objective examination revealed submandibular lymphadenopathy. Auscultation of the lungs revealed a slight crackling sound over the middle and lower parts of the lungs. The rest of the examination was unremarkable. Laboratory tests revealed elevated C-reactive protein (268 mg/L), interleukin-6 (1036 pg/mL), procalcitonin (182.7 ng/mL), lactate (2.2 mmol/L), alanine aminotransferase (112.05 UI/L), and aspartate aminotransferase (220.05 UI/L). Otherwise, the laboratory examination was unremarkable. The results of laboratory tests, vital signs, and signs and symptoms at the time of presentation are shown in Table 1. Chest X-ray showed interstitial pulmonary oedema. Abdominal ultrasound was unremarkable. The participant was not treated with antibiotics prior to presentation. The oropharyngeal swab was positive for the GAS antigen using a rapid point-of-care test. The patient was admitted to the ICU and given empiric antimicrobial therapy (penicillin 5 million units every 6 h). Based on the clinical presentation and the positive GAS antigen, STSS was suspected and clindamycin was added to the therapy at a dose of 600 mg every 8 h. The patient also underwent fluid resuscitation with 30 mL of a balanced crystalloid solution per kilogram of body weight within fast 3 h. The patient received conventional oxygen therapy due to dyspnoea and hypoxaemia. The patient responded well to fluid resuscitation (resolution of hypotension and tachycardia), and vasopressor therapy was not required. The patient was oliguric for the first 6 h, passing only 30 mL of urine. After the first 6 h and fluid replacement therapy, oliguria also resolved. On the second day of treatment, the patient’s vital signs were stable, and procalcitonin, neutrophilia, and lactate decreased. On the third day of treatment, the exanthema became lighter in colour, but desquamation of the palmar and plantar skin developed. The patient also developed strawberry tongue signs (Figure 6) and later developed palmar desquamation. Vital signs were stable and the patient was transferred to the general ward. Oropharyngeal swabs were later cultured for *Streptococcus pyogenes,* and blood cultures (two out of two) also revealed a pure culture of *Streptococcus pyogenes.* The patient met the diagnostic criteria for a definitive case of STSS. The clinical condition gradually improved, antibiotic therapy was discontinued on day 10, and the patient was discharged.

### 2.3. Case 3

A 38-year-old previously healthy Caucasian woman (ethnicity—Slovak, West Slavic) presented to the Emergency Department of the Clinic of Infectious Diseases and Geographical Medicine, University Hospital in Bratislava, with a four-day history of fever up to 40 degrees Celsius, progressive weakness, orthostatic collapses, pruritic rash, sore throat, nausea, and diarrhoea. On presentation, she was hypotensive (85/52 mm of mercury), tachycardic (110 beats per minute), and tachypnoeic (25 breaths per minute). The physical examination revealed a generalised bright red maculopapular rash with a sandpaper texture (Figure 7), deep red and swollen tonsils with purulent exudate, and submandibular lymphadenopathy. The physical examination was otherwise unremarkable. Laboratory investigations revealed elevated C-reactive protein (127 mg/L), interleukin-6 (438 pg/mL), procalcitonin (3.3 ng/mL), lactate (2.1 mmol/L), urea (8 mmol/L), and creatinine (128 umol/L). It also revealed mild hyponatremia (133 mmol/L) and hypoalbuminemia (27.2 g/L). The blood count revealed severe neutrophilia (22,310 cells/mL). The results of laboratory tests, vital signs, and signs and symptoms at the time of presentation are shown in Table 1. Chest X-ray revealed mild interstitial oedema. Abdominal ultrasound was unremarkable. The patient was admitted to the intensive care unit. Empiric antimicrobial therapy (ceftriaxone 2 g every 24 h) and fluid resuscitation (30 mL of a balanced crystalloid solution per kilogram of body weight within the first 3 h) were started. The oropharyngeal swab revealed antigen positivity of GAS by means of a rapid point-of-care test. Based on clinical presentation and the positive GAS antigen, STSS was suspected, and clindamycin, at a dose of 600 mg every 8 h, was added to therapy. Despite fluid resuscitation, the patient remained oliguric for the first 24 h, passing only 70 mL of urine. The next day, the patient developed dyspnoea and mild hypoxemia (oxygen saturation in the arterial blood at 92%); auscultation revealed crepitations over the lungs and chest X-ray revealed incipient interstitial oedema. Therefore, conventional oxygen therapy via nasal cannula was started. After 24 h of treatment, the vital functions stabilised, and oliguria, hypotension, and tachycardia resolved. Oropharyngeal swabs later revealed the culture of *Streptococcus pyogenes*. One of the three blood cultures taken at the time of presentation revealed a pure culture of *Streptococcus pyogenes*. On day 5, hypoxemia resolved, and the patient was transferred to the general ward. The rash became pale and only slightly apparent, and their sore throat, diarrhoea, and general weakness also resolved. Palmar desquamation evolved at day 7. Antibiotic therapy was discontinued on day 14, and the patient was discharged. 

## 3. Discussion

We present the case series of three patients with STSS with characteristic clinical presentation and positive point-of-care antigen tests for *Streptococcus pyogenes*. STSS is a rare disease. The aetiological agent responsible for this syndrome is almost exclusively *Streptococcus pyogenes* or GAS. STSS can rarely be caused by other group B, C, and G streptococci, such as *S. dysgalactiae subsp. equisimilis* or *S. agalactiae* [10,11]. It usually complicates invasive GAS soft tissue infections such as wound infections, cellulitis, or necrotizing fasciitis. In about half of cases, the skin lesion suggestive of a GAS infection entry site is obvious [1,2,8,9,10,11]. However, in more than 40% of cases, there is no obvious primary site of infection and the presence of GAS is only detected in the baseline blood culture [12,13]. Tonsillitis is a rare cause of STSS [2]. In our cases, sore throat and enlarged tonsils with purulent discharge were among the initial presentations, and GAS was isolated from the pharynx; there were no symptoms and signs of soft tissue infection. Thus, in these cases, STSS developed as a complication of tonsillitis. STSS is a serious and often fatal clinical condition. The pathophysiology of the disease is complex and not fully understood. However, it is known to be mediated by streptococcal toxins with superantigenic activity and interaction with the host immune system. Streptococcal exotoxins are also responsible for symptoms that overlap with scarlet fever, such as strawberry tongue and maculopapular rash with a sandpaper texture [4,5].

Superantigens activate T lymphocytes non-specifically by binding to major histocompatibility complex class II and T cell receptors [6]. The excessive non-specific activation of T cells leads to the abrupt release of pro-inflammatory cytokines, causing vasodilation and capillary leakage. The subsequent decline in perfusion leads to multi-organ dysfunction and failure. Eleven streptococcal toxins can act as superantigens. They are also known as streptococcal pyrogenic exotoxins [4,5]. Genes encoding these toxins are present in the genomes of all GAS strains. Even in potentially toxigenic strains, these genes are only expressed in the presence of specific environmental factors. Therefore, the complex interaction of GAS with its human host is required to trigger the development of STSS [7]. 

According to CDC diagnostic criteria, the diagnosis of STSS is based on the isolation of GAS and arterial hypotension, defined by a systolic blood pressure of 90 mmHg or less in adults and evidence of multi-organ involvement. It is characterised by two or more of the following: renal dysfunction, coagulopathy, liver involvement, acute respiratory distress syndrome, a generalised erythematous macular rash that may desquamate, soft tissue necrosis including necrotizing fasciitis or myositis, or gangrene. For the confirmed case, the condition of isolation of GAS from the typically sterile site, such as blood or cerebrospinal fluid, must be met. The case is probable if GAS is isolated from a non-sterile site [8]. The diagnosis in patient 1 can be concluded as probable STSS, as the GAS was not cultured from a sterile site. The patient fulfilled the diagnostic criteria for multi-organ involvement (acute respiratory distress syndrome, hypotension, renal impairment). However, based on the clinical presentation and the absence of other causes for the patient’s condition, we can conclude with a high degree of certainty that STSS was the cause of the patient’s illness. Patient 2 had a positive blood culture for GAS and met the diagnostic criteria for multi-organ involvement (acute respiratory distress syndrome, hypotension, acute renal impairment, acute liver impairment). Patient 3 had a positive blood culture for GAS and met the diagnostic criteria for multi-organ involvement (acute respiratory distress syndrome, hypotension, tachycardia, acute renal failure). Patients 2 and 3 also presented with typical clinical signs, namely a red maculopapular rash with a sandpaper texture, strawberry tongue, and palmar desquamation.

Initial symptoms are usually non-specific. They typically include flu-like symptoms (fever, chills, myalgia, headache) and non-specific gastrointestinal symptoms (nausea, vomiting). These symptoms precede the development of hypotension and signs and symptoms of multi-organ failure by 24 to 48 h [8]. In cases when STSS originates from soft tissue infection, severe limb pain may also be the initial symptom. In rare cases of STSS associated with tonsillitis, sore throat and enlarged tonsils with purulent discharge are usually present. [10]. The main risk factors for the development of STSS are age over 65 years, impaired skin integrity, and the presence of severe comorbidities such as alcohol abuse, diabetes mellitus, oncological disease, chronic obstructive pulmonary disease, and obesity. However, young and previously healthy individuals may also be affected [9,10,11]. In our cases, the patients were under 40 years of age. One had a history of heroin abuse, was obese, and was diagnosed with type 2 diabetes during hospitalisation. The other two patients were previously healthy. The lethality of STSS is very high, ranging from 30 to 70% [8]. Without prompt administration of effective antibiotics and supportive care, STSS rapidly progresses to multi-organ failure and is often fatal. Therefore, rapid administration of appropriate therapy is essential for successful management of the disease. Given the rarity of STSS and its potentially life-threatening consequences, it is essential for emergency physicians to be proficient in recognizing its symptoms. Due to the non-specific nature of the initial symptoms and the possibility that more definitive signs, such as the rash, may not be prominently visible upon presentation, a heightened level of vigilance is necessary for the prompt diagnosis of STSS in the emergency department. The diagnostic algorithm for STSS begins with the recognition of its signs and symptoms, followed by the localisation of the infection source and the acquisition of microbiological evidence of *Streptococcus pyogenes* infection. As bacteraemia is frequently present, obtaining blood cultures is considered the standard procedure. In the uncommon instances where STSS arises from tonsillitis, a swab from the tonsils is recommended [8]. The rapid test for the antigen of GAS may help us to find evidence of *Streptococcus pyogenes* infection faster than by using culture and may be useful in cases of STSS associated with tonsillitis. It has a relatively high sensitivity and acceptable specificity and is of high clinical value in the management of tonsillitis [14,15]. Sekizuka et al. described a case of STSS and pharyngitis in which a rapid test for the GAS antigen was also positive at the time of admission [2]. Therefore, these tests may help to diagnose STSS in cases with shock exanthema and sore throat, which may help to make the diagnosis quickly and administer appropriate treatment without delay. The main strength is that the results are available in a short period of time. The limitations are as follows: First, definitive diagnosis of STSS requires isolation of GAS from a sterile site, so the rapid antigen test cannot make a definitive diagnosis. Second, the rapid test does not provide information on the susceptibility of GAS to antimicrobial agents.

Treatment of STSS consists of supportive care for shock (fluid resuscitation, vasopressors, supplemental oxygen, renal replacement therapy if needed) and antimicrobial treatment, which should include high-dose beta-lactams and lincosamides. Penicillin remains the antimicrobial of choice for the treatment of GAS. Cephalosporins are also effective [16,17,18,19,20,21] and may be an alternative to penicillin in cases of penicillin allergy. This is particularly true of third-generation cephalosporins such as ceftriaxone and cefotaxime, which have a different R1 side chain to penicillin [22]. Lincosamides (clindamycin is the most commonly used), although bacteriostatic, are often used to treat severe streptococcal infections. They block the 50S subunit of the bacterial ribosome and inhibit the translation of streptococcal toxins. Therefore, the combination of lincosamides with penicillin is recommended [16,17].

There is evidence that adding clindamycin to beta-lactam therapy improves the prognosis of STSS. Notably, GAS resistance to clindamycin and macrolides is increasing [18,19,20]. This makes them unsuitable for monotherapy in STSS. Linezolid, which also inhibits photosynthesis, is very effective in treating severe GAS infections and could be used in the treatment of STSS [21]. Although penicillin alone will eventually stop toxin release, the combination of clindamycin or linezolid with penicillin results in a much faster decline in GAS toxin production than with penicillin alone [16]. The fact that clindamycin or linezolid should be added to treatment if STSS is suspected underlines the importance of early diagnosis of this disease. Therefore, point-of-care methods for GAS antigen assessment may improve the management of STSS by reducing the time delay between presentation and administration of appropriate antimicrobial treatment. 

This case series has several limitations. First, one of the patients (case patient 1) cannot be definitively classified as a case of STSS due to the absence of GAS cultivation from a sterile site. Second, standardised diagnostic and treatment algorithms were not employed, resulting in variations in the diagnostic and therapeutic approaches used among the case patients. Third, we were unable to provide the images of the *Streptococcus pyogenes* cultures as the cultures were provided by a third-party laboratory.

## 4. Conclusions

In conclusion, STSS is a rare complication of GAS tonsillitis. It is a serious and potentially fatal condition with significant mortality, and prompt diagnosis, prompt antimicrobial therapy, and supportive therapy are essential for survival. A point-of-care antigen test for GAS antigen may be helpful in promptly diagnosing tonsillitis associated with STSS.

## Figures and Tables

**Figure 1 clinpract-15-00070-f001:**
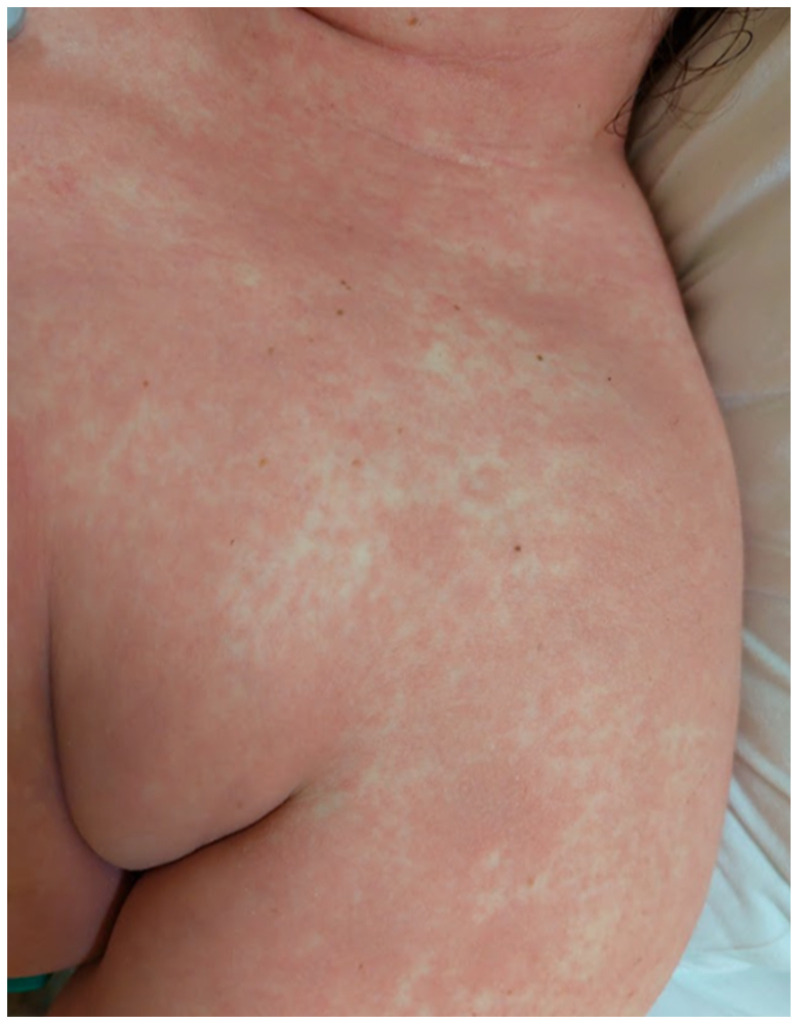
Exanthema over the left arm in patient 1. On presentation, the exanthema presented as a maculopapular, partially confluent, salmon-pink rash with a sandpaper texture.

**Figure 2 clinpract-15-00070-f002:**
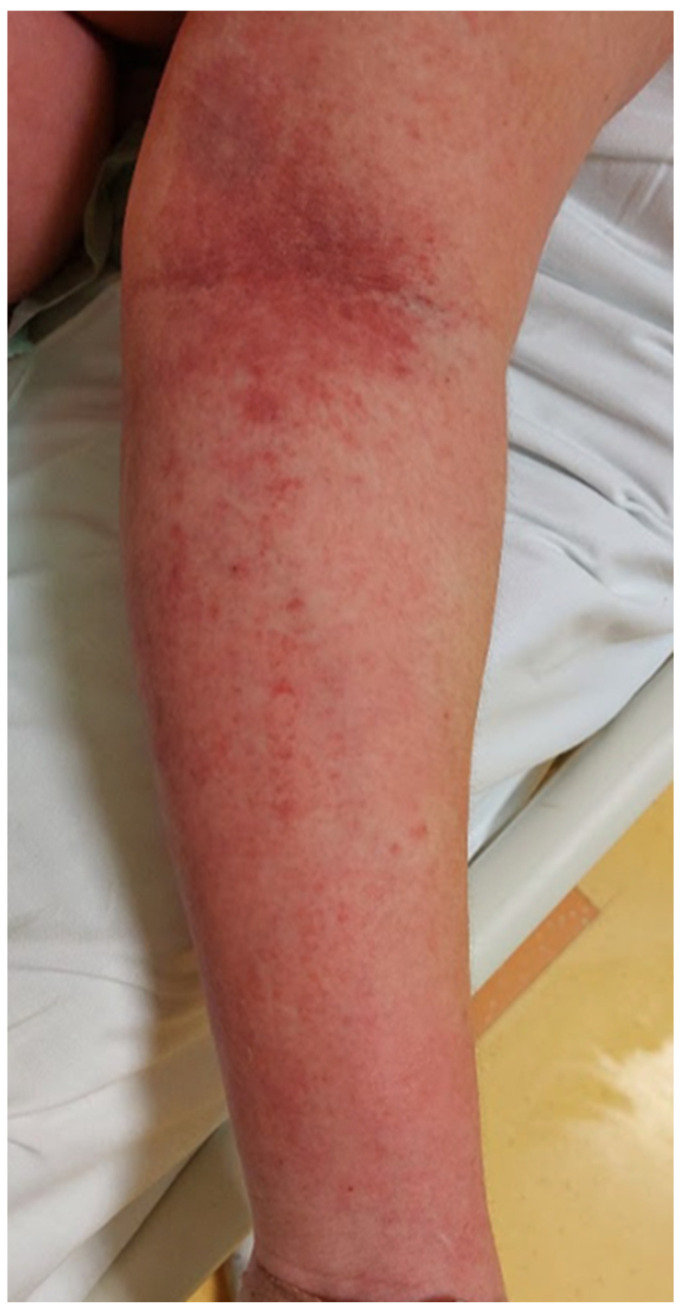
Exanthema over the cubital area and left forearm in patient 1. On presentation, the exanthema presented as a maculopapular, partly confluent, salmon-pink rash with a sandpaper texture. In the cubital area, the exanthema was darker red.

**Figure 3 clinpract-15-00070-f003:**
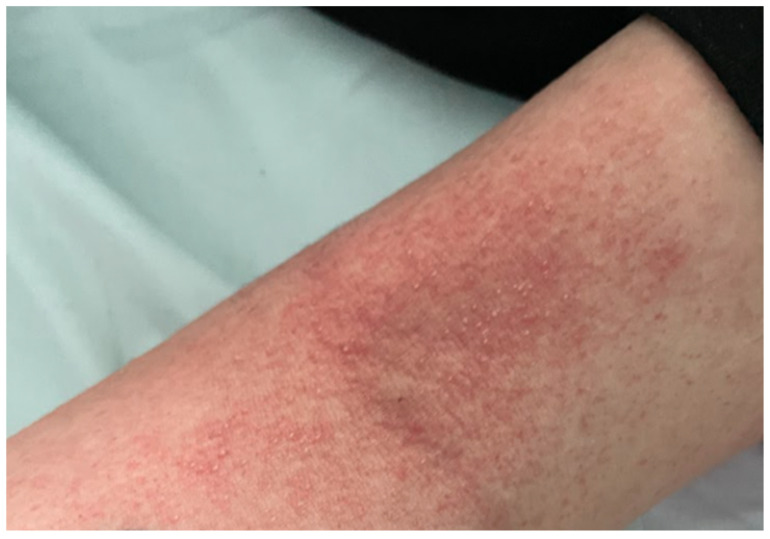
Exanthema over the cubital area in patient 2. On presentation, the exanthema presented as a maculopapular salmon-pink rash with a sandpaper texture.

**Figure 4 clinpract-15-00070-f004:**
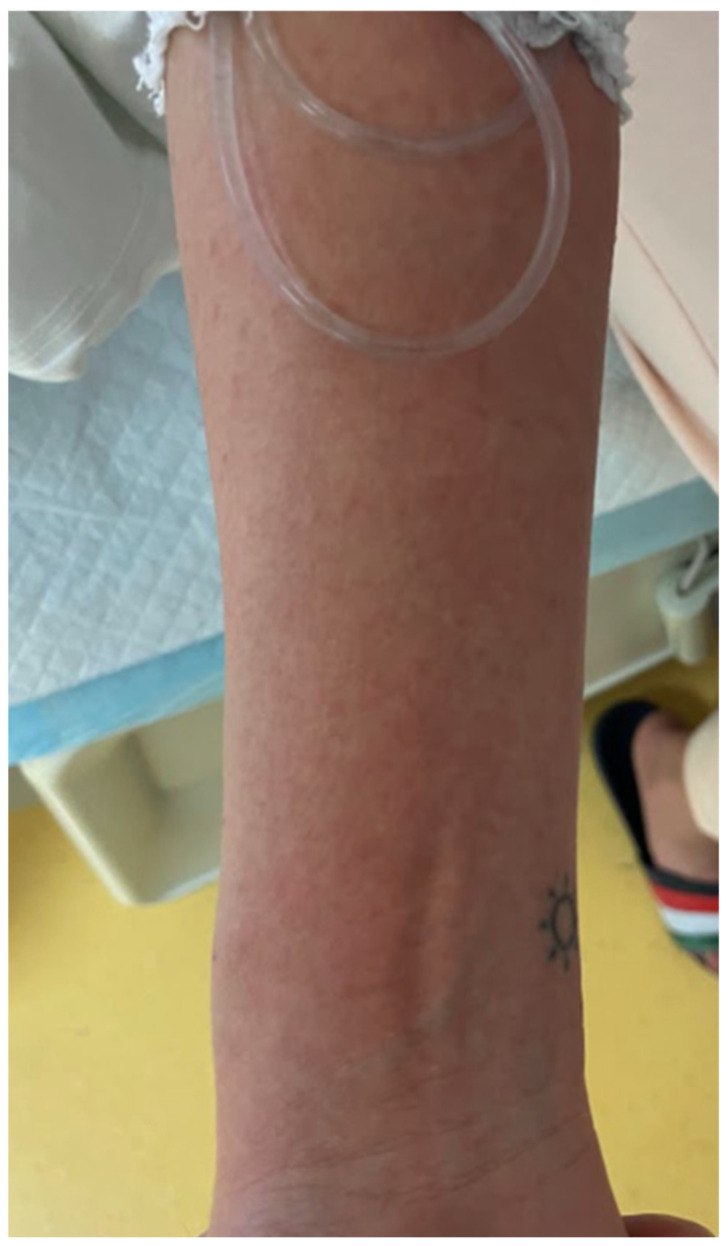
Exanthema on the right forearm in patient 2. On presentation, the exanthema presented as a maculopapular salmon-pink rash with a sandpaper texture.

**Figure 5 clinpract-15-00070-f005:**
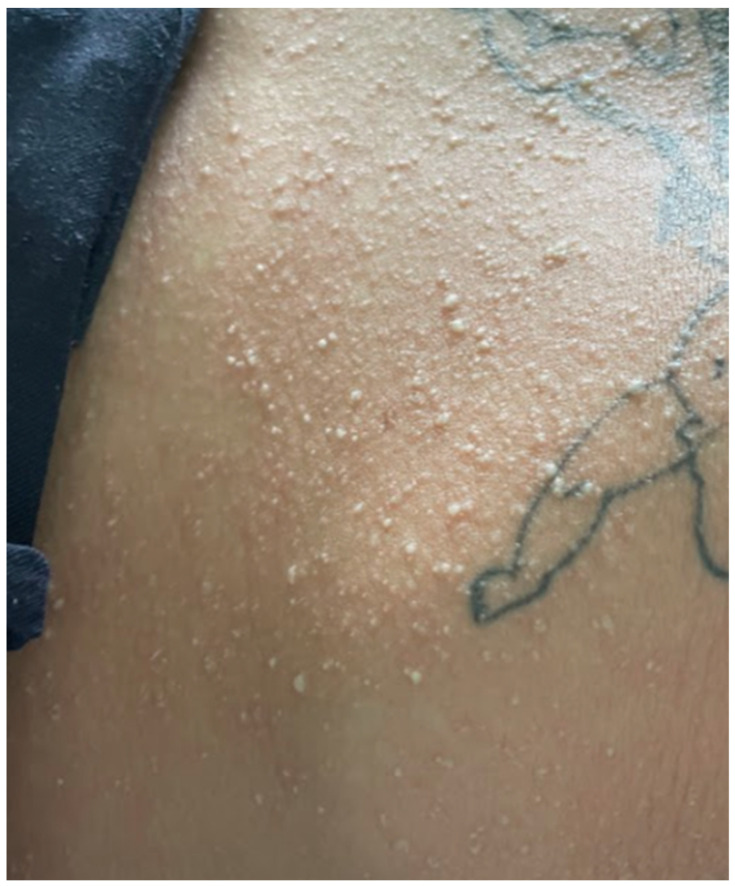
Abdominal exanthema in patient 2. On presentation, the exanthema appeared as a maculopapular rash with a sandpaper-like texture. The exanthema was paler in the abdominal area than on the extremities.

**Figure 6 clinpract-15-00070-f006:**
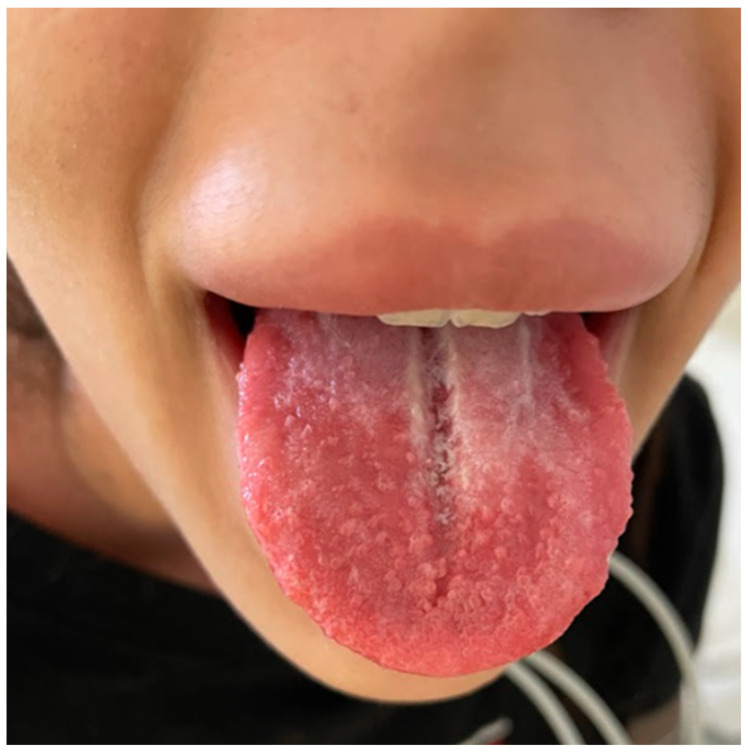
The sign of strawberry tongue in the case of patient 2.

**Figure 7 clinpract-15-00070-f007:**
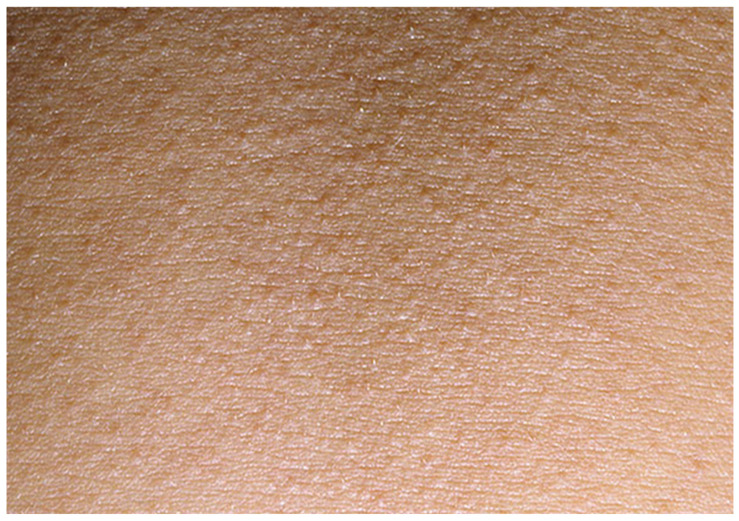
A close-up of the exanthema (abdominal area) in patient 3. The papules have the typical “sandpaper” appearance.

**Table 1 clinpract-15-00070-t001:** The vital signs, symptoms, signs, and results of the laboratory tests at the time of presentation.

Vital Signs	Patient 1	Patient 2	Patient 3
Systolic blood pressure (mmHg)	60	88/50	85
Diastolic blood pressure (mmHg)	30	50	52
Heart rate (beat/minute)	160	137	110
Symptoms and signs			
Dyspnoea	Present	Present	Present
Fever	Present	Present	Present
Rash	Present	Present	Present
Strawberry tongue	Present	Present	Present
Laboratory results			
C-reactive protein (mg/L)	321	268	127
Procalcitonin (ng/mL)	29.4	182.7	3.3
Creatinine (umol/L)	280	82	128
Interleukin-6 (pg/mL)	996	1036	438
Lactate (mmol/L)	3.9	2.2	2.1
Neutrophil count (cells/mL)	24,700	9500	22,310

## Data Availability

The data can be made available on reasonable request.

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
