# Peer review of "Streptococcal Toxic Shock Syndrome: Diagnostic and Therapeutic Approaches Incorporating Point-of-Care Antigen Testing—Case Series"

_clinpract, 2025, doi:10.3390/clinpract15040070_

Round 1

Reviewer 1 Report

Comments and Suggestions for Authors Toxic Shock Syndrome (TSS) is a high-risk life-threatening condition that affects the general population. However, individuals who have undergone surgery with catheters or prostheses, those hospitalized, or those with soft tissue infections—particularly those infected with Streptococcus pyogenes or Staphylococcus aureus—are more susceptible to this condition. This manuscript presents a clinical report describing three unrelated cases of TSS, two of which were associated with Streptococcal infection. In all cases, the diagnosis was made before the bacteria were isolated, and the rapid identification of the condition led to a satisfactory prognosis for the patients. Each case is meticulously described in the manuscript, focusing on the patients' symptoms. It is important to note that this work is a case report rather than original research, and therefore, it should not be evaluated as such. The contribution of this work lies in outlining the actions physicians should take when encountering patients with high fever, cutaneous rash, dyspnea, and hypotension.  

Author Response

Dear reviewer,

Thank you for your thoughtful review of our case series. We appreciate your insights and feedback regarding the presentation and implications of our findings.

We acknowledge your clarification regarding the classification of our work as a case report rather than original research. We appreciate your recognition of the careful description of the three unrelated cases of Toxic Shock Syndrome (TSS) and the emphasis on the rapid identification and management of this condition.

In response to your comments, we have made the following revisions to enhance clarity and impact:

  1. Emphasis on Clinical Actions: We have further emphasized the critical actions physicians should take upon encountering patients with symptoms indicative of TSS.

  2. Clarification of Diagnosis Algorithm: We added a more detailed section (in the discussion) on the diagnostic algorithm, highlighting the importance of early recognition in improving patient outcomes.

Reviewer 2 Report

Comments and Suggestions for Authors

COMMENTS TO THE AUTHORS

The authors present an interesting and important case-series featuring the use of point-of-care antigen test in the diagnosis of tonsillitis complicated by STSS. The main message of the study is that an early diagnosis of STSS can be established by using a rapid antigen test. Therefore, the patients could receive appropriate antibiotic therapy, which saved their life. 

It is interesting to note that the first patient had drug abuse, overweight and diabetes as predisposing factors, while the other patients did not have any of it.

I only have a few comments and questions to the authors.

1.     Is the study reported by the ‘CARE Guideline’ for case studies? If yes, state it in the manuscript and upload the Checklist, please.

2.     It would be interesting to see a table showing the changes in the patients’ lab test during hospitalization, please provide a table with the laboratory results.

3.     State a strength and limitation section of the study.

Author Response

Daer reviewer

Thank you for your thoughtful review of our case report. We appreciate your insights and recommendations.

  1. Is the study reported by the ‘CARE Guideline’ for case studies? If yes, state it in the manuscript and upload the Checklist, please. - We followed the CARE guideline, we added the statement about the guidelines in the statement section and uploaded the CARE checklis
  2. It would be interesting to see a table showing the changes in the patients’ lab test during hospitalization, please provide a table with the laboratory results. - We added the table with the laboratory results and signs and symptoms of the patients. Page 11. 
  3. State a strength and limitation section of the study. - We stated the limitations in the discussion section. Page 13.

Reviewer 3 Report

Comments and Suggestions for Authors

The title "Point-of-care antigen test for rapid diagnosis of streptococcal toxic shock - case series" couldn't cover the main text. The whole text described the case including diagnosis and treatment. However, the authors described the core part of point-of-care antigen test too little. So, the title need to be changed. If the authors kept the title, they should emphasize on the test,  point-of-care antigen test. Moreover, the pathogen culture pictures also be provided. 

Author Response

Daer reviewer

Thank you for your thoughtful review of our case report. We appreciate your insights and recommendations.

  1. The title "Point-of-care antigen test for rapid diagnosis of streptococcal toxic shock - case series" couldn't cover the main text. The whole text described the case including diagnosis and treatment. However, the authors described the core part of point-of-care antigen test too little. So, the title need to be changed. If the authors kept the title, they should emphasize on the test,  point-of-care antigen test. - 

    In response to your comments regarding the title, we have amended it to more accurately reflect the content of our manuscript. The revised title emphasizes both the diagnostic and therapeutic approaches utilized in the cases presented, while still incorporating the aspect of point-of-care antigen testing, as you suggested. We believe this change enhances clarity and better captures the primary focus of the case series.

    Additionally, we have taken care to elaborate on the diagnostic and treatment protocols provided in the manuscript to ensure comprehensive coverage of the key aspects of our findings.

    The new title is: 

    Streptococcal Toxic Shock Syndrome: Diagnostic and Therapeutic Approaches Incorporating Point-of-Care Antigen Testing - Case Series

  2. Moreover, the pathogen culture pictures also be provided. - Unfortunately, we could not provide the pictures of the pathogens culture. An external commercial lab provides the cultures and were unavailable to the authors. We stated this limitation in the discussion section. 

Reviewer 4 Report

Comments and Suggestions for Authors

Title:   Point-of-care antigen test for rapid diagnosis of streptococcal toxic shock - case series 
The authors present how point-of-care testing for GAS antigens aided in rapid diagnosis of STSS in 3 cases of tonsilitis. The rapid diagnosis of STSS and timely administration of antibiotics, along with supportive care, are crucial for providing better treatment and a favourable outcome. The authors used a rapid antigen test, which enabled them to administer a prompt antibiotic regime to the patient compared to waiting for the results of a bacterial culture and identifying the bacteria and the antigen. The manuscript addresses the topic of rapid diagnostics and their impact on clinical outcomes, which is both original and highly relevant to the field of healthcare. The authors have addressed a significant gap by highlighting how the timely diagnosis can influence patient treatment and overall health outcomes. This study will add a crucial dimension to our understanding of healthcare delivery by demonstrating that rapid diagnostics not only facilitate timely treatment but also improve patient outcomes.  I recommend this manuscript be accepted after minor revisions.
My minor comments are as follows:
Line 9: Italicise the Genus and species names throughout the manuscript.
Use justify alignment 
Line 30: Reference 11 is before references 8,9, and 10 in Line 32. Please organise the references in order throughout the manuscript.
Line 49: correct ‘houdr’ to ‘hour’.
Lines 63,68,69 70: Use either ‘h’ or ‘hour’. Use uniform notation throughout the manuscript. 
Line 73: Correct ‘priaor’ to ‘prior’.
Line 74: Correct ‘tonisles’ to ‘tonsils’. 
Line 78: Rephrase the sentence for clarity.
Line 152: Do the authors mean diagnostic criteria?
Line 237: Figure 7- Do the authors mean sandpaper appearance?
Line 249: Add a period to the sentence ‘Tonsillitis is a rare cause of STSS [2]’.
Authors can provide a summary of the symptoms in the three cases and the time taken to recover/be discharged from the Hospital. 

Comments on the Quality of English Language

The manuscript has many incorrect spellings. Minor grammatical editing is also required. 

Author Response

Dear reviewer, 

We greatly appreciate your input and recommendations. We believe that our manuscript will benefit greatly from the revisions you have suggested.

Minor comments are as follows:
Line 9: Italicise the Genus and species names throughout the manuscript. - We Italicise the genus and species names
Line 30: Reference 11 is before references 8,9, and 10 in Line 32. Please organise the references in order throughout the manuscript. - We rearanged the references to be in right order.
Line 49: correct ‘houdr’ to ‘hour’. - We corrected the spelling
Lines 63,68,69 70: Use either ‘h’ or ‘hour’. Use uniform notation throughout the manuscript. - We decided to use "hour" in whole manuscript
Line 73: Correct ‘priaor’ to ‘prior’. - We corrected the spelling
Line 74: Correct ‘tonisles’ to ‘tonsils’. - We corrected the spelling
Line 78: Rephrase the sentence for clarity. - We rephrased the sentence for more clarity
Line 152: Do the authors mean diagnostic criteria? - We changed the "criteria" to "diagnostic criteria"
Line 237: Figure 7- Do the authors mean sandpaper appearance? - We rewrite the figure title to be more clarifying. We mend sandpaper appearance. 
Line 249: Add a period to the sentence ‘Tonsillitis is a rare cause of STSS [2]’. - We added period.
Authors can provide a summary of the symptoms in the three cases and the time taken to recover/be discharged from the Hospital.  - We added the table sumarising the symptoms, signs and results of laboratory tests. Time to discharge is stated in the case reports as the last sentence. We would rather not include this parameter to the table to make the table of initial symptoms as clear and informative as possible. 

- We corrected the spelling errors (Great Britain EnglisH)

Round 2

Reviewer 3 Report

Comments and Suggestions for Authors

The revised manuscript has been greatly improved.  I'd like to recommend publication in the Journal.